Subject Areas:
ecology, health and disease and epidemiology

Keywords:
diarrhoeal disease, precision health mapping

Author for correspondence:
Michelle V. Evans
e-mail: mvevans@uga.edu

# Socio-demographic, not environmental, risk factors explain fine-scale spatial patterns of diarrhoeal disease in Ifanadiana, rural Madagascar

Michelle V. Evans[1,2], Matthew H. Bonds[4,5,6], Laura F. Cordier[5,6], John M. Drake[1,2], Felana Ihantamalala[4,5,6], Justin Haruna[5,6], Ann C. Miller[4], Courtney C. Murdock[1,2,3,7], Marius Randriamanambtsoa[8], Estelle M. Raza-Fanomezanjanahary[9], Bénédicte R. Razafinjato[5,6] and Andres C. Garchitorena[5,6,10]

[1]Odum School of Ecology, [2]Center for Ecology of Infectious Diseases, and [3]Department of Infectious Diseases, College of Veterinary Medicine, University of Georgia, Athens, GA, USA
[4]Department of Global Health and Social Medicine, Blavatnik Institute at Harvard Medical School, Boston, MA, USA
[5]PIVOT, Ranomafana, Madagascar
[6]PIVOT, Boston, MA, USA
[7]Department of Entomology, College of Agriculture and Life Sciences, Cornell University, Ithaca, NY, USA
[8]National Institute of Statistics, Antananarivo, Madagascar
[9]Ministry of Health, Antananarivo, Madagascar
[10]MIVEGEC, Univ. Montpellier, CNRS, IRD, Montpellier, France

MVE, 0000-0002-5628-0502; JMD, 0000-0003-4646-1235; CCM, 0000-0001-5966-1514

Precision health mapping is a technique that uses spatial relationships between socio-ecological variables and disease to map the spatial distribution of disease, particularly for diseases with strong environmental signatures, such as diarrhoeal disease (DD). While some studies use GPS-tagged location data, other precision health mapping efforts rely heavily on data collected at coarse-spatial scales and may not produce operationally relevant predictions at fine enough spatio-temporal scales to inform local health programmes. We use two fine-scale health datasets collected in a rural district of Madagascar to identify socio-ecological covariates associated with childhood DD. We constructed generalized linear mixed models including socio-demographic, climatic and landcover variables and estimated variable importance via multi-model inference. We find that socio-demographic variables, and not environmental variables, are strong predictors of the spatial distribution of disease risk at both individual and commune-level (cluster of villages) spatial scales. Climatic variables predicted strong seasonality in DD, with the highest incidence in colder, drier months, but did not explain spatial patterns. Interestingly, the occurrence of a national holiday was highly predictive of increased DD incidence, highlighting the need for including cultural factors in modelling efforts. Our findings suggest that precision health mapping efforts that do not include socio-demographic covariates may have reduced explanatory power at the local scale. More research is needed to better define the set of conditions under which the application of precision health mapping can be operationally useful to local public health professionals.

# 1. Introduction

Over 700 000 child deaths are attributed to diarrhoeal disease (DD) annually [1]. The burden of DD is unequally distributed across the globe: 73% of deaths occur in just 15 low-income countries, driven by inequalities in water and sanitation infrastructure and environmental conditions [2]. Understanding the risk factors of spatio-temporal patterns of DD can be instrumental in designing public health interventions that target populations most at risk. Precision health mapping is an approach that incorporates increasingly available fine-scale social and environmental information into spatial models to explain and predict spatial disease patterns at resolutions finer than those previously possible [3]. This approach has recently emerged as a method to identify areas and populations at risk of disease and has been successfully used to map the global distribution of diseases with strong environmental signatures, such as malaria [4] and schistosomiasis [5]. There are few examples, however, of efforts that are precise enough and appropriately integrated with implementation to be able to improve local public health strategies.

DD is ideal for precision health mapping because the determinants of environmental suitability for a diarrhoeal pathogen, such as climate [6,7] and land cover [8], are well known. Hydrological networks, as well as infrastructure and water, sanitation and hygiene (WASH) practices influence transmission dynamics and individual risk of DD [8,9]. Upstream land cover has been shown to predict DD prevalence in rural areas of the tropics [8], with cumulative effects for populations that are downstream of sources of water contamination (e.g. livestock or agricultural run-off [10]).

These studies tend to rely on data extracted from large national surveys, such as Demographic and Health Surveys, that are powered to estimate indicators at broad spatial and temporal scales (e.g. country or region every 5 years). When projected to more granular geospatial data, they produce fitted values on the assumption that relationships found at broad spatial scales exist at fine spatial scales. However, because these localized predictions are not typically fitted from data at fine spatio-temporal scales, they may fail to explain patterns of disease at those scales, limiting their ability to inform priorities set by local health actors. For example, several recent studies analysed the spatial patterns of disease and healthcare access across Africa, incorporating datasets that included GPS-tagged individuals and clusters [11–13]. While these continent-level studies are comprehensive in their coverage across countries, the finest resolution dataset used for Madagascar (Demographic and Health Surveys 2008 surveys) had an average resolution of one cluster per approximately 1000 km$^2$. To improve local disease control, health managers in Madagascar make decisions at very small administrative levels (Fokontany, average size 34 km$^2$ in Madagascar) much finer than global or continent-wide studies are intended to address.

Localized analysis matters because socio-ecological determinants of DD may be region- and pathogen-specific; relationships identified at the global level may not hold locally. For example, despite a proven biological pathway of faecal–oral transmission, the effects of WASH interventions on DD prevalence are ambiguous owing to differences in socio-ecological context and specific aetiological agents [14]. Pathogens' temperature responses differ between viral (higher survival in colder environments) and bacterial agents (lower growth in colder environments) [15], with net effects on DD dynamics dependent on the prevailing set of pathogens. At the national and global scale, increased forest cover is consistently associated with lower incidence of DD [8,16,17], but these associations have not been tested at the fine spatial scale relevant for local action. Addressing these inherent challenges in downscaling precision health mapping tools is central to evaluating their use.

Here, we investigated the potential for precision health mapping of DD at a fine-scale (village-level) in the rural health district of Ifanadiana, in southeastern Madagascar. We used multiple spatio-temporal datasets, including a district-representative longitudinal cohort study and health centre case reports in Ifanadiana, to identify the socio-ecological variables associated with DD. We then assessed our ability to predict disease risk at a scale relevant to public health managers. The district comprises a protected tropical rainforest, large areas of agricultural land and a steep east–west elevation gradient. Owing to widespread poverty and low access to improved WASH infrastructure, the population has high exposure to diarrhoeal pathogens, with children under the age of five and infant mortality rates at twice the national estimates [18,19]. Because of the data availability, the region's environmental variability and the population's high exposure rates, Ifanadiana should be well-suited for the application and validation of precision health mapping at a local scale.

# 2. Methods

## (a) Study area

Ifanadiana district encompasses an area of 3970 km$^2$ in the Vatovavy-Fitovinany region of southeastern Madagascar and consists of about 210 000 people across 13 communes (the second smallest administrative unit). The majority of the district is rural and the dominant land cover is agricultural for rice production. The western border includes Ranomafana National Park, a 416 km$^2$ protected tropical rainforest. The east–west elevation gradient slopes downwards from an altitude of 1400 m in the western border to less than 100 m in the eastern border. This combination of densely forested area, agricultural fields and human-developed lands offers an ideal study system to explore how human-modified landcover influences diarrhoeal pathogen transmission.

Beginning in 2014, the non-governmental organization PIVOT partnered with the Ministry of Health (MoH) to establish a district-level model health system [20]. The intervention strengthens the public health system through a set of programmes focused on improving system readiness (infrastructure, personnel, equipment, supply chain), clinical programmes (maternal and child health, emergency care, infectious diseases) and integrated information systems at all levels of the district health system (community health, primary care centres and hospital). Initially started in four communes in 2014, the intervention has since reached six of 13 communes (as of 2019) and will expand to the whole of the district by the end of 2020. In addition, a pilot initiative is underway to upgrade the community health system to support professionalized proactive community case management [21], in which community health workers visit each household monthly and are able treat DD, malaria and respiratory infections for children under five to reduce geographical accessibility challenges. Central to this agenda is optimizing the interventions geographically in the context of heterogeneous disease burdens; the work described in this study was done in support of these initiatives.

## (b) Data collection

Our analysis included two datasets on childhood DD: a longitudinal cohort study from a representative sample of 1600

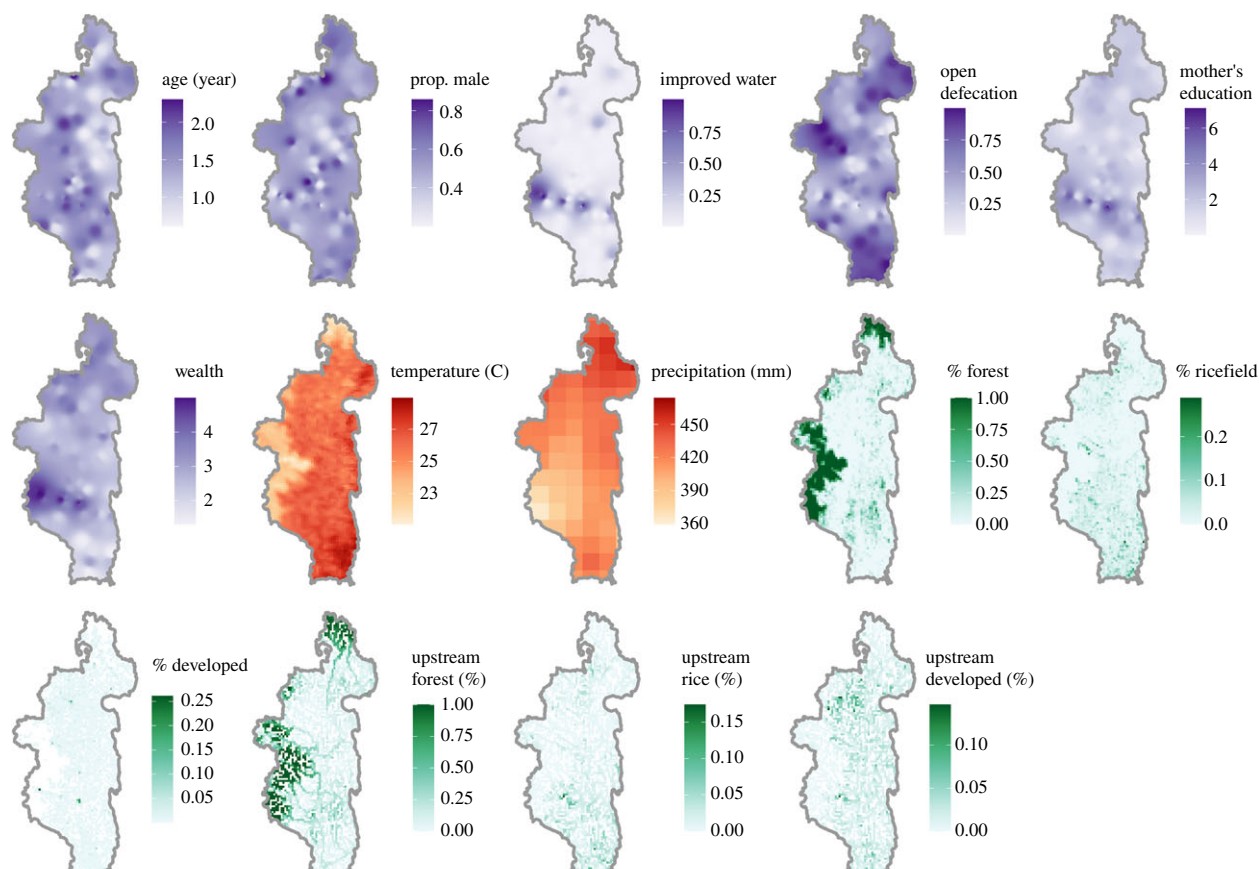

**Figure 1.** Socio-ecological variables used in the prevalence model mapped across Ifanadiana district. Only data for 2014 is shown. Socio-demographic variables (purple) were interpolated from 80 cluster centroids using third-order inverse distance weighting. Climate variables (red) are shown in their initial spatial resolution. Landcover variables (green) were spatially averaged to 1 × 1 km pixels for visualization. (Online version in colour.)

households aggregated into 80 clusters in Ifanadiana at 2 year intervals (April/May 2014, August/September 2016 and April/May 2018) [18] and monthly diarrhoeal incidence from public primary care centres from 13 communes in Ifanadiana from June 2015 to December 2018. The longitudinal cohort study included socio-demographic variables at the individual and household level and individual-level DD positivity (electronic supplementary material, table S1, see the electronic supplementary material, appendix for further detail on each variable). A child was determined positive for DD if diarrhoeal symptoms had been reported in the two weeks prior to the survey date.

Climate variables were collected from temporally explicit remotely sensed imagery (electronic supplementary material, table S1), and aggregated to the spatial resolution of the cohort cluster or administrative commune, depending on the analysis. We explored including mean monthly temperature, monthly precipitation and cumulative precipitation up to six months prior, but owing to high collinearity between variables we ultimately chose one temperature and one precipitation metric to represent climate over the prior six months based on each covariates' centrality to the other metrics [22]. The final variables chosen were temperature at a two-month lag and precipitation over the prior three months for the cohort data analysis and temperature at a one-month lag and cumulative rainfall over the prior three months for the health system analysis. These variables should be interpreted as representative of temperature or precipitation over the prior six months to the survey date or health system report because they were all similarly explanatory. Landcover variables were derived from OpenStreetMap and Sentinel-2 satellite imagery (see the electronic supplementary material, appendix for detailed methods). A large portion of households surveyed in the cohort obtained their water from surface water sources compared to other water sources (46%), and surface water is vulnerable to contamination from

upstream pollutants [10]. Therefore, we calculated the percentage of each type of upstream landcover for streams within 1 km of a surveyed village as a covariate.

## (c) Analysis: disease prevalence using the cohort dataset

We used multi-model inference to identify socio-ecological variables associated with individual child's positivity for DD, or disease prevalence. Multi-model inference allows multiple hypotheses to be considered and incorporates measurements of parameter uncertainty through the process of model averaging [23]. Unlike stepwise approaches that result in one final model, this technique results in a final set of models, as determined by information criteria such as Akaike's information criteria (AIC), which are used to calculate parameter estimates and their uncertainty. Importantly, this approach is well-suited for the study of socio-ecological systems, often characterized by multiple, interacting variables, owing to its ability to identify variables that are consistently strong predictors of the response variable across models [24]. The initial global model was a generalized linear model with hierarchical random effects of the survey cluster nested within the year of survey, to account for the sampling design of the cohort survey study. A total of six individual or household-level socio-demographic variables and eight cluster-level environmental variables were included in the initial global model (figure 1; electronic supplementary material, table S1). Only main effects without interactions were included. All variables were inspected for assumptions of normality, and the six land cover variables were *ln*-transformed. Following Gelman [25], we centred and standardized our predictor variables to a mean of 0 with 0.5 standard deviation to allow for comparison of parameter estimates during model averaging. The response variable was

whether an individual child had an episode of DD in the prior two weeks, as reported by the mother (binary). Therefore, we used a binomial distribution with a logit link (logistic regression). To further assess the robustness of our findings, we also conducted out-of-sample testing. We included a randomly sampled subset of 56 of the 80 clusters in our training dataset, and assessed the model's performance predicting DD in the remaining 24 clusters. We then generated a full model set consisting of all possible subsets of the global model using the training data. We created a subset that included all models that fit as well as the best fitting models by including models with AIC scores within two AIC of the best fitting model, following established methods using information criteria [26]. An average model was obtained from this subset of top models using the zero method of model averaging [27], implemented in the AICcmodavg package [28] in R v. 3.5.2. We assessed model performance using the area under the receiver operator curve (AUC) and Tjur's coefficient of discrimination, $D$, which are performance metrics suitable for models where the response variable is binomial. AUC balances the true positive rate and false positive rate to return a metric ranging from 0 to 1, where a value of 0.5 is equal to a completely random model. Tjur's $D$ is the average difference of the predicted values for positive and negative cases and is an analogue of $R^2$, ranging from 0 to 1 [29]. The performance was assessed for both in-sample and out-of-sample datasets, accounting for fixed effects only.

## (d) Analysis: commune-level incidence using health system information

We used a similar multi-model inference approach as described above to explore the socio-ecological risk factors of monthly DD incidence at the level of the commune. The initial global model was fitted using a negative binomial distribution and included commune as a random intercept. More complex models including spatially structured random effects and autocorrelation were also considered, but either they offered no improvement on fit (as measured by AIC) over the model with only a random effect of commune, or they did not converge (electronic supplementary material, table S2). There was no evidence of temporal or spatial autocorrelation in model residuals. The model included a subset of the covariates used in the cohort survey analysis. Age and sex were not included because they varied at the level of the individual child and were not spatially structured (figure 1). Similarly, the upstream landcover variables were not included because communes are much larger than the watersheds delineated in our flow accumulation model, and so a spatially aggregated summary of such large areas could be misleading. Socio-demographic variables were aggregated to the commune-level by taking the mean of household-level variables from the three cohort surveys and were interpolated linearly to estimate monthly values from June 2015 to December 2018 (electronic supplementary material, table S1). Discussions with local health workers suggested that DD incidence increased following the national Independence Day (26 June), and so we included the month of July as a covariate in the model. This holiday is unique because it occurs after the main rice harvest period (March–May), when subsistence farmers in Ifanadiana have disposable income to spend on celebrations, resulting in an extraordinary consumption of meat products and fried foods. This sudden change in diet in combination with the lack of appropriate sanitary conditions to store these food products for days is locally hypothesized to drive the rise of diarrhoea in the days and weeks that follow. Finally, we included whether or not a commune was part of the PIVOT catchment during that month to control for increased health centre use following the health system strengthening activities implemented by the MoH and PIVOT in these areas [19]. We assessed out-of-sample performance by training the initial model on a random subset of 9 of

13 communes, stratified across the PIVOT catchment area. Out-of-sample performance was then assessed on the remaining four communes, two of which were in the PIVOT catchment area, accounting for fixed effects only. Because the response variable was incidence per commune, we used a different metric than in the first analysis to assess model performance: normalized root-mean square error (NRMSE), scaled by the standard deviation. This metric can be interpreted as the ratio of variation not explained by the model to the overall variation in DD incidence, with higher values representing a poorer model fit.

## 3. Results

### (a) Disease prevalence

The cohort survey dataset consisted of 2745 children under the age of five across three sampling periods. There were between four and 30 children in each of the 80 clusters (mean ± s.d.: $11.45 \pm 3.55$). District-wide prevalence of DD ranged from 9% to 19% across sampling periods (2014: 145/896, 2016: 127/850, 2018: 84/999), and there was little evidence for spatial correlation in prevalence across clusters (figure 2; Moran's I ranged from 0.00 to 0.06). Additionally, there was low correlation of cluster-level DD prevalence between years (Spearman's $\rho$: 2014/2016: 0.270, 2014/2018: 0.077, 2016/2018: 0.123).

The top model set consisted of 27 models out of 65 536 total models that were within two $\Delta$ AIC of the best fit model (electronic supplementary material, table S3). The top model set included all 14 of the original predictors, two of which (age and sex) were in all 27 top models (figure 3). A quadratic relationship better explained the relationship between age and DD than a linear relationship (figure 3c). The rate of disease increased from age one to two years, and then decreased with increasing age. A supplemental analysis investigating the effects of wealth separately for economic variables (e.g. asset, land and livestock ownership) and housing quality (e.g. flooring, roofing and wall materials) found that only economic variables were associated with the individual risk of DD, similar to the relationship with the wealth index in the main model (electronic supplementary material, figure S1). Male children were more likely to have had diarrhoea in the prior two weeks than female children (odds ratio = 1.55, 1.18–2.04 95% confidence interval (CI)). The regression coefficients of all environmental and climatic variables were included in less than half of the models, except for the proportion of rice fields, (figure 3), suggesting they were not important risk factors of DD. Finally, the fitted model performed poorly, with an in-sample AUC of 0.66 and Tjur's $D$ of 0.033 and out-of-sample AUC of 0.57 and Tjur's $D$ of 0.015.

### (b) Commune-level incidence

The monthly diarrhoeal incidence in children under 5 years of age ranged from 0 to 62 cases per thousand across all 13 communes, with a clear peak in incidence in the winter months (June–August) (figure 4). This seasonality was consistent across years. There were differences across communes as well, with the five PIVOT-supported communes consistently reporting the highest incidences (figure 4). Eight models out of 2048 were within two AIC of the top-performing model and were included in the averaged model (electronic supplementary material, table S4). The average model included

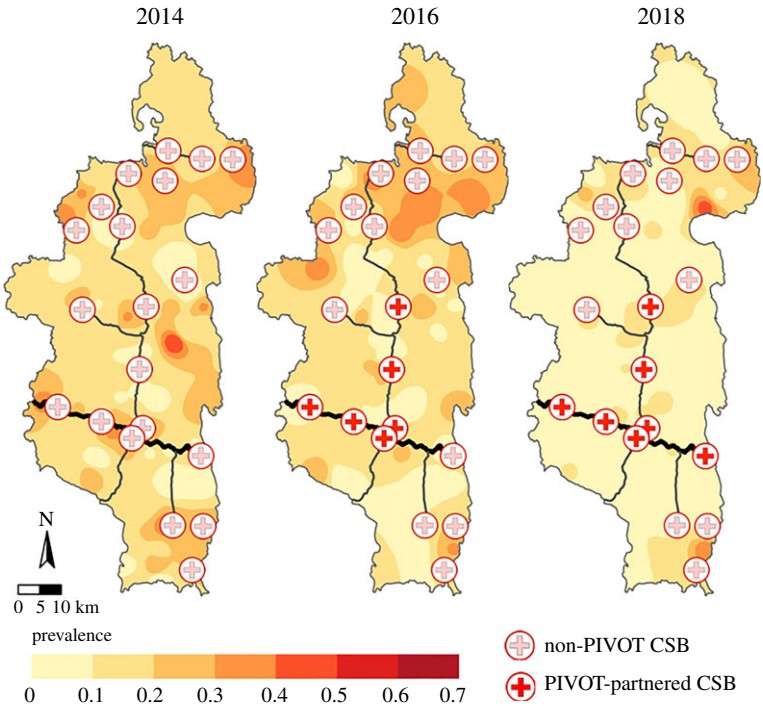

**Figure 2.** Diarrhoeal disease (DD) prevalence was not spatially structured across clusters. Maps illustrate childhood DD prevalence in prior two weeks interpolated from 80 cluster centroids using third-order inverse distance weighting across the three survey years. Public primary care centres (CSBs) are represented by the cross symbols (filled = PIVOT supported) and the major roads are drawn in black (thick = paved). (Online version in colour.)

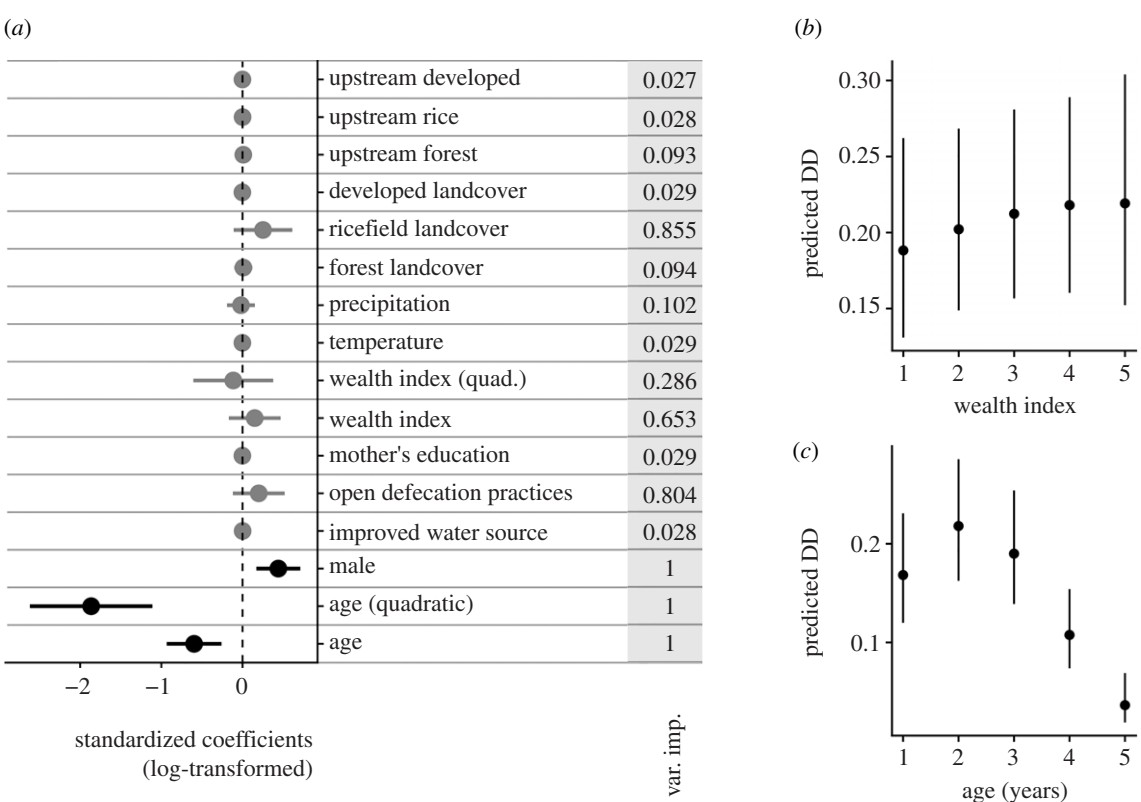

**Figure 3.** Coefficients from model predicting diarrhoeal disease (DD) prevalence from longitudinal cohort data. (*a*) Log odds-ratios from model predicting DD prevalence from the longitudinal cohort survey data and variable importance scores (in grey column). Variables not included in the final averaged model are not plotted, and variables only included in a subset of the final top model set are plotted in grey. Socio-demographic variables were included in all models and had coefficients with 95% confidence intervals (CIs) that did not overlap zero. (*b*) Predicted disease risk (mean and 95% CI) for children across wealth indices, holding all other variables constant. (*c*) Predicted disease risk (mean and 95% CI) for children across ages, holding all other variables constant.

nine of the 11 original covariates, with only rice landcover and open defecation not included in any model. Four covariates (occurrence of the national holiday, a commune falling in the PIVOT catchment, temperature and precipitation) were included in all models and all four had regression coefficients whose 95% CIs did not include zero (figure 5). We estimated a

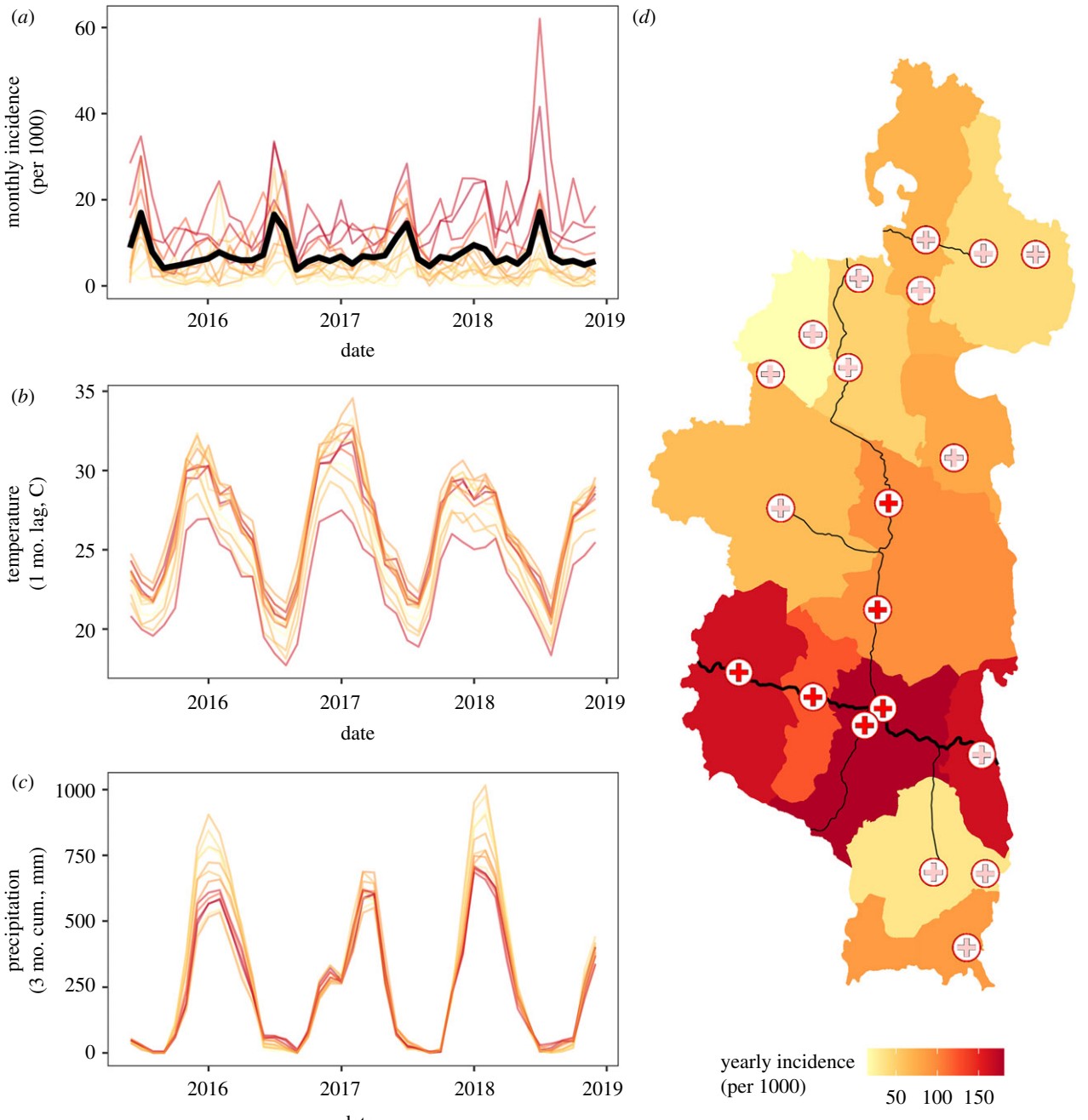

**Figure 4.** Commune-level diarrhoeal incidence in children under five across time and space. (*a*) Monthly diarrhoeal disease incidence for each commune, with the mean incidence across all communes plotted in the bold black line. (*b*) One-month lagged temperature values for each commune. (*c*) Three-month cumulative rainfall values for each commune. (*d*) Map of yearly incidence by commune for 2017. Health centres supported by PIVOT are red-filled symbols as in figure 2. Line colours in (*a*), (*b*) and (*c*) correspond to incidence values in (*d*). (Online version in colour.)

NRMSE of 0.65 on the in-sample dataset and 2.68 on the out-of-sample dataset. This suggests that the model was able to explain nearly half of the variance in the training data, but nearly three times the overall variation in DD incidence remained unexplained when applying the model to the testing data, indicating that the model was not able to adequately explain spatial patterns in DD.

Unlike the analysis of disease prevalence, both socio-economic and environmental variables influenced disease incidence. The relationship between DD incidence and wealth was weak and nonlinear, as reflected by the inclusion of the linear ($\beta = -0.20$, $-0.88$–$0.47$ 95% CI) and quadratic ($\beta = 0.37$, $-0.85$–$1.52$ 95%) wealth index terms in the final model set. Incidence was higher in communes where a

larger proportion of the population had access to an improved water source ($\beta = 0.55$, $-0.04$–$1.16$ 95% CI), although these CIs did overlap zero. Similarly, there was a trend for lower DD incidence in communes with higher proportions of developed landcover ($\beta = -0.64$, $-1.30$ to $0.006$ 95% CI). Temperature and precipitation influenced seasonal disease incidence, with lower temperatures and higher precipitation associated with higher incidence rates (figure 5; temperature $\beta = -0.42$, $-0.60$ to $-0.23$ 95% CI; rainfall $\beta = 0.33$, $0.15$–$0.517$ 95% CI). All other environmental variables had low variable importance scores (figure 5), suggesting they had little influence on disease incidence, and that the main contribution of environmental variables was in driving seasonality of disease incidence.

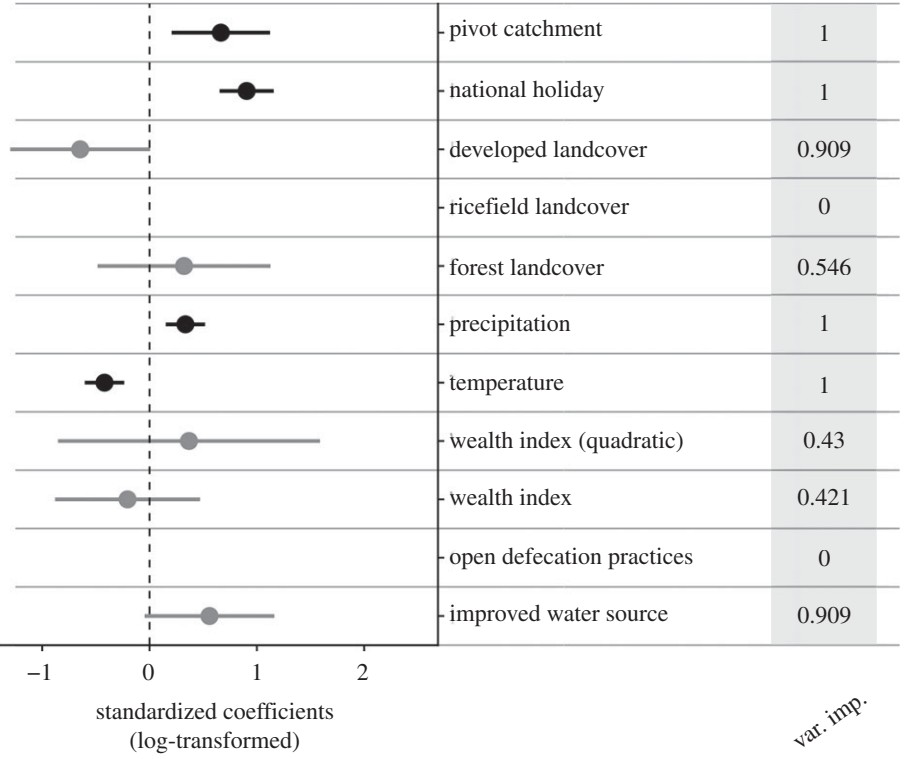

**Figure 5.** Coefficients from model predicting commune-level diarrhoeal disease incidence from health system data and variable importance scores (in grey column). Variables not included in the final averaged model are not plotted, and variables only included in a subset of the final top model set are plotted in grey.

## 4. Discussion

There is growing interest in the potential for 'big data' to be used to support increasingly optimized and proactive public health interventions. Predicting DD risk through precision health mapping accordingly offers great potential to prevent diarrhoea-caused morbidity and mortality. However, there is a substantial gap between our collective analytic capabilities and the use of those capabilities to solve problems where they matter the most—in areas of extreme poverty with high burdens of disease. For precision health mapping to lead to actionable interventions, the global relationships between DD and socio-ecological covariates must retain fine spatial granularity relevant to public health actors. Here, we used a multi-dataset analysis from a model health district to identify social and ecological predictors of diarrhoea in children under 5 years of age in rural Madagascar. Environmental variables contributed little to spatial variation in disease incidence and prevalence but did explain the seasonality of DD. Socio-demographic variables, however, were predictive of DD risk at both the individual and commune levels. This questions whether precision health mapping approaches that rely solely on environmental and climatic variables can accurately predict disease risk at local scales.

Results from both modelling exercises revealed inconsistent patterns in the relationship between wealth and DD risk. These findings are contrary to the general consensus that diarrhoeal prevalence and morbidity are lower in wealthier households [30], which may have access to improved WASH infrastructure and health care. However, these studies are often done at a national or multi-national scale, where wealth variability may be larger, and they include both urban and rural populations. Given that over 70% of the population in Ifanadiana lives in extreme poverty [18] and the highest commune-level coverage of access to an improved water source was only 53% in our dataset, it is possible that living conditions of households across the range of wealth scores were comparable, resulting in similar exposure to diarrhoeal pathogens. By contrast, individual-level variables of child sex and age explained a large portion of the variation in DD prevalence. In particular, the nonlinear trend in DD prevalence across age agrees with the results of a continent-wide study of childhood DD in Africa [31]. This age-dependent pattern may be owing to the implementation of a rotavirus immunization campaign in Madagascar since May 2014, which vaccinates children at two and four months of age [32]. Indeed, there was a drop in cohort-wide prevalence from 16.18% to 8.41% from 2014 to 2018 following the start of immunization campaigns; however, there was no evidence for a similar decrease in commune-level incidence rates across this time period (figure 4a). As immunization campaigns continue, the planned collection of paired data on rotavirus vaccination status and DD in Ifanadiana will allow further investigation.

We found evidence of very strong seasonality in DD incidence (figure 4a), which is in agreement with other studies focused on enteric diseases in the tropics [33,34]. DD incidence was higher during the cooler, drier winter months, and our analysis found cooler temperatures and higher precipitation to be associated with increased incidence. Enteric viruses have higher survival and environmental persistence at colder temperatures [35], which may suggest that enteric viruses are the primary causative agents of DD in Ifanadiana. This is further supported by a study conducted in human populations near Ranomafana National Park that found viral infection rates of over 50% during the austral winter [36]. Colder temperatures during winter months may also

be associated with human behaviour changes that increase transmission rates, such as indoor crowding. Regarding precipitation, virus persistence is higher in higher-moisture soils [15], and higher precipitation could increase exposure to enteric viruses by increasing soil moisture and thereby viral persistence.

We conclude that climatic factors do not contribute significantly to spatial variation in DD prevalence or incidence across Ifanadiana district, although this may be owing to the specific limitations of our data. Although the district spans an elevational gradient of 100–1400 m, its area is only approximately 45 × 125 km at its widest, which may not offer enough climate variation across space to observe a clear spatial trend or explain a significant amount of the variance in DD burdens, compared to that seen at national or multi-national scales, where environmental variation is greater. For precipitation especially, which is available at a 5 km resolution, there is little variation across clusters within a sampling period (figure 1). The collection of finer-scale rainfall or watershed-level discharge data could aid in exploring these relationships further, but unfortunately is not available for this geographical region. Another limitation of our data is the timing and frequency of the longitudinal cohort surveys in relation to the seasonality in DD. The surveys took place outside of the seasonal peak of DD. Reported DD cases were low in these months and there is less variation across communes (figure 4a), which resulted in weaker spatial patterns in DD for our model to fit.

After controlling for seasonal climate variables, the occurrence of the national holiday remains associated with higher DD incidence. Studies on short-term changes in disease dynamics owing to seasonal cultural events in low- and middle-income countries are rare, but changes in contact rates associated with holidays and mass migration have been shown to influence disease transmission elsewhere [37]. In Ifanadiana, the movement of people associated with the national holiday could increase contact rates and pressure on existing sanitation systems, leading to increased transmission of DD. Additionally, changes in diet and increased food sharing associated with celebration of the holiday could increase children's exposure to DD pathogens [38]. This national holiday is only one example of a cultural factor that may influence spatio-temporal patterns in disease burdens, and modelling efforts should strive to incorporate these local nuances through collaborations with local stakeholders and interdisciplinary teams that include social scientists [39].

Unlike past studies [8,10], we find little evidence for a spatial pattern in DD associated with nearby or upstream land use. Although it is hypothesized that undisturbed forest may filter contaminants from water and lower risk of DD for populations downstream [8], the largest forest in Ifanadiana, Ranomafana National Park, attracts thousands of tourists annually. In this case, the additional water contamination owing to tourism may negate any beneficial 'filtering' mechanisms of the forest. While the amount of developed area, a fine-scale proxy for population density, was associated with lower incidence rates of DD, this relationship was not significant. Ifanadiana is a primarily rural district, with only a few large towns (population greater than 3000 people) located along paved roads. Communes

with large towns along the main road did have higher incidence of DD in the health system dataset, however, this variation was not associated with the amount of developed area. Therefore, it is likely that unmeasured characteristics of these communes, such as mobility or water quality, are contributing to their high rates of DD incidence.

In conclusion, we fail to find evidence that precision health mapping accurately describes local patterns in DD prevalence and incidence rates in this context. Therefore, precision health mapping may not actually be useful for informing local health authorities of target areas for the implementation of public health interventions for DD at a fine scale. Despite the availability of extensive longitudinal epidemiological data and high-resolution landcover and environmental information, we found that our models poorly predicted disease prevalence. Precision health mapping at this scale may require even finer-scale spatial data, such as that collected via community health worker programmes that combine proactive care with mobile-health data collection. The strongest predictors of disease risk in our context are socio-demographic or cultural but their lack of spatial structure compared to environmental variables may limit the performance of precision health mapping exercises. Precision health mapping should be developed and adapted for different environmental and social contexts in order to better define the set of conditions under which its application at fine spatial scales can be of use to public health professionals.

**Ethics.** The original longitudinal study was reviewed and approved by the Madagascar National Ethics Committee and the Harvard Medical School IRB, and de-identified data were provided by the authors for the current study. The current analysis was ruled non-human subjects work by the University of Georgia IRB.

**Data accessibility.** Environmental data and code to reproduce the analysis and figures are available on figshare (https://dx.doi.org/10.6084/m9.figshare.13061744). De-identified health and socio-demographic data are available upon reasonable request (research@pivotworks.org) due to IRB regulations.

**Authors' contributions.** M.V.E. contributed to the conceptualization of the project, data curation, analysis, methodology and drafting and editing the manuscript. M.H.B. contributed to the conceptualization of the project, data curation, analysis, methodology and drafting and editing the manuscript. L.F.C. contributed to data curation, project administration and management and editing the manuscript. J.M.D. contributed to the conceptualization of the project, methodology and drafting and editing the manuscript. F.I. contributed to data curation, methodology and editing the manuscript. J.H. contributed to data curation and project administration and management. A.C.M. contributed to data curation and editing the manuscript. C.C.M. contributed to drafting and editing the manuscript. M.R. contributed to data curation. E.M.R.-F. contributed to editing the manuscript. B.R.R. contributed to data curation. A.C.G. contributed to the conceptualization of the project, data curation, analysis, methodology, project administration and management and drafting and editing the manuscript.

**Competing interests.** We declare no competing interests.

**Funding.** M.V.E. was supported by an NSF Graduate Research Fellowship while conducting this research.

**Acknowledgements.** We would like to thank the PIVOT research interns, Mauricianot Randriamijaha, Tanjona Andreambeloson and Miadana Rakotozafinirainy, for their contribution to the OpenStreetMap data and support on this project. We also thank the PIVOT community team and health centre staff for their help in the collection of the health data and three anonymous reviewers for their comments and edits to the manuscript.

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
