## [Peer Review File · Proceedings of the Royal Society B: Biological Sciences]

Review History

RSPB-2020-1018.R0 (Original submission)

Review form: Reviewer 1

Recommendation

Accept with minor revision (please list in comments)

Scientific importance: Is the manuscript an original and important contribution to its field?

Good

General interest: Is the paper of sufficient general interest?

Excellent

Quality of the paper: Is the overall quality of the paper suitable?

Good

Is the length of the paper justified?

Yes

Should the paper be seen by a specialist statistical reviewer?

No

Do you have any concerns about statistical analyses in this paper? If so, please specify them explicitly in your report.

No

It is a condition of publication that authors make their supporting data, code and materials available - either as supplementary material or hosted in an external repository. Please rate, if applicable, the supporting data on the following criteria.

Is it accessible?

N/A

Is it clear?

N/A

Is it adequate?

N/A

Do you have any ethical concerns with this paper?

No

Comments to the Author

The manuscript by Evans et al attempts to identify the relative importance of climatic, socio-demographic and landcover on spatio-temporal diarrhea incidence at microscale level. While the results reveal new insights about the relative importance of these factors at a local scale, they did not discuss in details the limitation of the data used and its impact on the conclusion. I think the challenge of getting a long term data at the local scale will always limit the application of precision health mapping. Overall, the manuscript is well written and the manuscript would have greater scientific significance if the concerns detailed below are addressed by the authors.

Main concerns

1. My first concern is the use of cohort data to build the model. Diarrhea in highly seasonal in Ifanadiana (Fig. 4A) with peak in winter months (June – August) and building models using longitudinal cohort study with 2 month measurements per year with 2-year intervals may likely lead to a bias results. For instance, April/May and August/September are outside the peak months and I think this contributed to the low impact of the climate variables particularly rainfall in the overall model. May be the authors can discuss this as a potential limitation to the study.
2. Another concern is the failure of the authors to justify the reason for using only rainfall at 1 month lag and 6 months accumulation rainfall. While the authors gave some reasons for using temperature at 1 month lag temperature in the supplementary materials, I think it will be prosper for the authors to do same for rainfall. May be some exploratory analysis should first be carried out as studies have found association of different indices of rainfall with diarrhea.

Minor issues and typos

1. Fig 1. - The values precipitation (lag -1) looks a bit strange (the values are too small). It is more like temperature values. Please I suggest the authors to cross check it.

2. Table 1. - I think upstream forest should be under landcover to make it consistent with statement on line 177

3. Line 68 - transmission dynamics and individual risk of DD(10)(12,13). Please put the references together.

4. Line 232 - within 2 Δ AIC of the best fit model (Table S1) --- I think the Table reference should be S2 instead of S1.

5. Line 257 - (Table S2). The average model included eleven of the twelve original covariates. Please Table reference should be S3 instead of S2.

Review form: Reviewer 2

Recommendation

Major revision is needed (please make suggestions in comments)

Scientific importance: Is the manuscript an original and important contribution to its field?

Acceptable

General interest: Is the paper of sufficient general interest?

Acceptable

Quality of the paper: Is the overall quality of the paper suitable?

Acceptable

Is the length of the paper justified?

Yes

Should the paper be seen by a specialist statistical reviewer?

Yes

Do you have any concerns about statistical analyses in this paper? If so, please specify them explicitly in your report.

No

It is a condition of publication that authors make their supporting data, code and materials available - either as supplementary material or hosted in an external repository. Please rate, if applicable, the supporting data on the following criteria.

Is it accessible?

No

Is it clear?

N/A

Is it adequate?

N/A

Do you have any ethical concerns with this paper?

No

Comments to the Author

Please see attached file.

Decision letter (RSPB-2020-1018.R0)

20-Jul-2020

Dear Ms Evans:

I am writing to inform you that your manuscript RSPB-2020-1018 entitled "Socio-demographic, not environmental, risk factors explain fine-scale spatial patterns of diarrheal disease in Ifanadiana, rural Madagascar" has, in its current form, been rejected for publication in Proceedings B.

This action has been taken on the advice of referees, who have recommended that substantial revisions are necessary. With this in mind we would be happy to consider a resubmission, provided the comments of the referees are fully addressed. However please note that this is not a provisional acceptance.

Sincerely,
Dr Sasha Dall
mailto: proceedingsb@royalsociety.org

Associate Editor
Comments to Author:

I have now received two referees reports. As you can see both referees are positive about the work, but also raise a number of detailed issues including questions of cross-validation. I think the general question addressed is an interesting one and found the paper well written and presented. However, you need to address the substantial criticisms either through further analysis or a detailed response.

Reviewer(s)' Comments to Author:

Referee: 1

Comments to the Author(s)

The manuscript by Evans et al attempts to identify the relative importance of climatic, socio-demographic and landcover on spatio-temporal diarrhea incidence at microscale level.

While the results reveal new insights about the relative importance of these factors at a local scale, they did not discuss in details the limitation of the data used and its impact on the conclusion. I think the challenge of getting a long term data at the local scale will always limit the application of precision health mapping. Overall, the manuscript is well written and the manuscript would have greater scientific significance if the concerns detailed below are addressed by the authors.

Main concerns

1. My first concern is the use of cohort data to build the model. Diarrhea is highly seasonal in Ifanadiana (Fig. 4A) with peak in winter months (June - August) and building models using longitudinal cohort study with 2 month measurements per year with 2-year intervals may likely lead to a bias results. For instance, April/May and August/September are outside the peak months and I think this contributed to the low impact of the climate variables particularly rainfall in the overall model. Maybe the authors can discuss this as a potential limitation to the study.

2. Another concern is the failure of the authors to justify the reason for using only rainfall at 1 month lag and 6 months accumulation rainfall. While the authors gave some reasons for using temperature at 1 month lag temperature in the supplementary materials, I think it will be prosper for the authors to do same for rainfall. Maybe some exploratory analysis should first be carried out as studies have found association of different indices of rainfall with diarrhea.

Minor issues and typos

1. Fig 1. - The values precipitation (lag -1) looks a bit strange (the values are too small). It is more like temperature values. Please I suggest the authors to cross check it.

2. Table 1. - I think upstream forest should be under landcover to make it consistent with statement on line 177

3. Line 68 - transmission dynamics and individual risk of DD(10)(12,13). Please put the references together.

4. Line 232 - within 2 Δ AIC of the best fit model (Table S1) --- I think the Table reference should be S2 instead of S1.

5. Line 257 - (Table S2). The average model included eleven of the twelve original covariates. Please Table reference should be S3 instead of S2.

Referee: 2

Comments to the Author(s)

Please see attached file.

Author's Response to Decision Letter for (RSPB-2020-1018.R0)

See Appendix A.

RSPB-2020-2501.R0

Review form: Reviewer 1

Recommendation

Accept with minor revision (please list in comments)

Scientific importance: Is the manuscript an original and important contribution to its field?

Excellent

General interest: Is the paper of sufficient general interest?

Good

Quality of the paper: Is the overall quality of the paper suitable?

Good

Is the length of the paper justified?

Yes

Should the paper be seen by a specialist statistical reviewer?

No

Do you have any concerns about statistical analyses in this paper? If so, please specify them explicitly in your report.

No

It is a condition of publication that authors make their supporting data, code and materials available - either as supplementary material or hosted in an external repository. Please rate, if applicable, the supporting data on the following criteria.

Is it accessible?

Yes

Is it clear?

Yes

Is it adequate?

Yes

Do you have any ethical concerns with this paper?

No

Comments to the Author

The manuscript by Evans et al presents new insights about potential drivers of local-scale diarrhea incidence. While I recommend the manuscript to be published, I suggest few major issues that needs to be addressed by the authors.

Major issues

1. A first major issue with the manuscript is the interpretation and conclusion regarding holiday (national Independence Day) as a significant predictor. I agree that holidays may increase diarrhea cases, however, drawing a major conclusion based on a single holiday event maybe misleading. In addition, June 26 (or July used in the model) is within peak season of transmission.

I suggest the authors to support this conclusion by looking at whether diarrhea cases also increase after other holidays.

2. The authors failed to interpret the patterns of predicted disease risk for children across ages (Fig. 3C). A potential explanation might be due to rotavirus vaccination provided it has been introduced in the population. I suggest the authors examine these more closely and discuss the Fig. 3C.

Minor issues and typos

254 - sampling periods. There were between four and 30 children in each of the 80 clusters. I suggest authors to be consistent regarding four and 30.

265 - then decreased with increasing age.. Delete one full stop.

Decision letter (RSPB-2020-2501.R0)

29-Dec-2020

Dear Ms Evans:

Your manuscript has now been peer reviewed and the reviews have been assessed by an Associate Editor. The reviewers' comments (not including confidential comments to the Editor) are included at the end of this email for your reference. As you will see, the reviewers have raised some concerns with your manuscript and we would like to invite you to revise your manuscript to address them.

Research ethics:

Use of animals and field studies:

It is a condition of publication that you make available the data and research materials supporting the results in the article (<https://royalsociety.org/journals/authors/author-guidelines/#data>). Datasets should be deposited in an appropriate publicly available repository and details of the associated accession number, link or DOI to the datasets must be included in the Data Accessibility section of the article (<https://royalsociety.org/journals/ethics-policies/data-sharing-mining/>). Reference(s) to datasets should also be included in the reference list of the article with DOIs (where available).

If you wish to submit your data to Dryad (<http://datadryad.org/>) and have not already done so you can submit your data via this link [http://datadryad.org/submit?journalID=RSPB&manu=\(Document not available\)](http://datadryad.org/submit?journalID=RSPB&manu=(Document%20not%20available)), which will take you to your unique entry in the Dryad repository.

Please submit a copy of your revised paper within three weeks. If we do not hear from you within this time your manuscript will be rejected. If you are unable to meet this deadline please let us know as soon as possible, as we may be able to grant a short extension.

Best wishes,

Dr Sasha Dall
mailto:proceedingsb@royalsociety.org

Reviewer(s)' Comments to Author:
Referee: 1

Comments to the Author(s).

The manuscript by Evans et al presents new insights about potential drivers of local-scale diarrhea incidence. While I recommend the manuscript to be published, I suggest few major issues that needs to be addressed by the authors.

Major issues

1. A first major issue with the manuscript is the interpretation and conclusion regarding holiday (national Independence Day) as a significant predictor. I agree that holidays may increase diarrhea cases, however, drawing a major conclusion based on a single holiday event maybe misleading. In addition, June 26 (or July used in the model) is within peak season of transmission. I suggest the authors to support this conclusion by looking at whether diarrhea cases also increase after other holidays.
2. The authors failed to interpret the patterns of predicted disease risk for children across ages (Fig. 3C). A potential explanation might be due to rotavirus vaccination provided it has been introduced in the population. I suggest the authors examine these more closely and discuss the Fig. 3C.

Minor issues and typos

254 - sampling periods. There were between four and 30 children in each of the 80 clusters. I suggest authors to be consistent regarding four and 30.

265 - then decreased with increasing age.. Delete one full stop.

Author's Response to Decision Letter for (RSPB-2020-2501.R0)

See Appendix B.

Decision letter (RSPB-2020-2501.R1)

05-Feb-2021

Dear Ms Evans

I am pleased to inform you that your manuscript entitled "Socio-demographic, not environmental, risk factors explain fine-scale spatial patterns of diarrheal disease in Ifanadiana, rural Madagascar" has been accepted for publication in Proceedings B.

You can expect to receive a proof of your article from our Production office in due course, please check your spam filter if you do not receive it. PLEASE NOTE: you will be given the exact page

length of your paper which may be different from the estimation from Editorial and you may be asked to reduce your paper if it goes over the 10 page limit.

Open Access

Paper charges

Sincerely,

Dr Sasha Dall

Appendix A

1785

The University of Georgia

®

October 7, 2020

Re: Manuscript No. RSPB-2020-1018

Dear Editor:

Please find enclosed with this letter our revised version of our manuscript “Socio-demographic, not environmental, risk factors explain fine-scale spatial patterns of diarrheal disease in Ifanadiana, rural Madagascar” which we resubmit for publication as a Research Article in *Proceedings of the Royal Society B*.

Your comments and those of the reviewers were insightful and informative and greatly enhanced the quality of our manuscript. In the following pages are our responses to each of the comments.

Reviewers’ comments are written in sans-serif font, with our replies directly following in *italics*. Revisions in the text are shown using Track Changes, and line numbers in the replies refer to this document. We hope that the revisions and our accompanying responses will be sufficient to make our manuscript suitable for publication in *Proceedings of the Royal Society B*.

I can be reached by email (mvevans@uga.edu), phone (703 725 9580), or post (Odum School of Ecology, University of Georgia, Athens GA 30602). Thank you for your consideration.

Sincerely,
Michelle Evans

We confirm that this manuscript is original and has not been published, is not in press, and is not under review elsewhere. All R code needed to reproduce the analysis will be deposited on Figshare upon acceptance and can currently be accessed via the following private link: <https://figshare.com/s/fc9cc786d9d6f9b7f84a>.

Reviewer 1:

The manuscript by Evans et al attempts to identify the relative importance of climatic, socio-demographic and landcover on spatio-temporal diarrhea incidence at microscale level. While the results reveal new insights about the relative importance of these factors at a local scale, they did not discuss in details the limitation of the data used and its impact on the conclusion. I think the challenge of getting a long term data at the local scale will always limit the application of precision health mapping. Overall, the manuscript is well written and the manuscript would have greater scientific significance if the concerns detailed below are addressed by the authors.

Main concerns

1. My first concern is the use of cohort data to build the model. Diarrhea is highly seasonal in Ifanadiana (Fig. 4A) with peak in winter months (June – August) and building models using longitudinal cohort study with 2 month measurements per year with 2-year intervals may likely lead to biased results. For instance, April/May and August/September are outside the peak months and I think this contributed to the low impact of the climate variables particularly rainfall in the overall model. Maybe the authors can discuss this as a potential limitation to the study.

Thank you for mentioning this. The district-wide surveys are meant to collect data on a variety of health and wellness indicators and so are not designed to capture the peak of the diarrheal disease seasonality. This is indeed an important limitation to our study, and we have added discussion of this to our manuscript [LINES 537 - 542]:

“Another limitation of our data is the timing and frequency of the longitudinal cohort surveys in relation to the seasonality in DD. The surveys took place in April/May and August/September, outside of the seasonal peak of DD in June-August. Reported DD cases in the health system were low in these months and there is less variation across communes (Fig. 4A), which resulted in weaker spatial patterns in DD for our model to fit.”

2. Another concern is the failure of the authors to justify the reason for using only rainfall at 1 month lag and 6 months accumulation rainfall. While the authors gave some reasons for using temperature at 1 month lag temperature in the supplementary materials, I think it will be proper for the authors to do the same for rainfall. Maybe some exploratory analysis should first be carried out as studies have found association of different indices of rainfall with diarrhea.

In response to comments from both reviewers, we chose to adjust how collinearity was addressed in our analysis. We had originally explored lags for temperature and precipitation from 1 to 6 months, as well as accumulation over the previous 2 to 6 months. Then, we used the pre-scan method of selecting variables based on their

performance in univariate regressions of the response variable following Dormann et al. 2013, thus selecting for multivariate regressions with the variable for temperature and rainfall that was most associated with DD, to avoid collinearity. However, we do not want this to be interpreted as implying that the temperature and precipitation values at other lags are not predictive, as they were all highly correlated ($\rho > 0.9$) and so would have similar predictive capacities. Therefore, again following Dormann et al. 2013, we have chosen the most central variable of each group of climatic variables to represent the full range of temporal lags that were considered a priori. In the case of precipitation, this resulted in a lagged precipitation and cumulative precipitation value that were still correlated with $\rho > 0.7$. We therefore re-calculated the variable centrality measure across lagged and cumulative rainfall, resulting in one measure to represent precipitation patterns over the prior six months. This process was repeated separately for the model using cohort data and the model using health system data. The final variables chosen were temperature at a two month lag and precipitation over the prior three months for the cohort data analysis and temperature at a one month lag and cumulative rainfall over the prior three months for the health system analysis. This is described in the main text on LINES 195-205 and in the supplement.

Minor issues and typos

1. Fig 1. - The values precipitation (lag -1) looks a bit strange (the values are too small). It is more like temperature values. Please I suggest the authors to cross check it.

We have updated this figure to represent the new variables included in the cohort-model. This value is indeed in mm. The prior figure was illustrating precipitation from April of 2014, which is the end of the rainy season and has little precipitation. This has now been changed to cumulative precipitation over the prior three months to match our variable collinearity selection process.

2. Table 1. - I think upstream forest should be under landcover to make it consistent with statement on line 177

We have grouped both climate and landcover variables under the umbrella category "Environmental Variables". Unfortunately, the way the table is split across pages in the word document makes this difficult to read, but we do not think this will cause a problem once typeset. We've also adjusted the margins of the Table document to fix this in the word document.

3. Line 68 - transmission dynamics and individual risk of DD(10)(12,13). Please put the references together.

Thank you for the sharp eye. We have fixed this typo.

4. Line 232 - within 2 Δ AIC of the best fit model (Table S1) --- I think the Table reference should be S2 instead of S1.

5. Line 257 - (Table S2). The average model included eleven of the twelve original covariates. Please Table reference should be S3 instead of S2.

Yes, thank you for pointing these two typos out. We have adjusted them.

Reviewer 2:

Abstract

Line 29: precision health mapping can map the general geospatial distribution of health indicators. Hotspots imply that the technique is mostly suitable to discerns areas of significantly unique disease burden which is not entirely true

Line 31: Many precision health mapping projects use data that is GPS-tagged and therefore not collected at “coarse scales” (see work from Simon Hay’s group)

Line 41: The sentence “Our findings ... “ does not follow from the previous sentences. Is it implied that global precision mapping efforts use primarily environmental variables? If so, please refer to the many precision health mapping papers such Mosser et al 2016 that use more than environmental variables in their efforts. While the result may hold perhaps the attack on global precision health mapping efforts may need to be softened

We have softened our language throughout the abstract to focus on the benefits of including fine-scale data sources and socio-demographic data, rather than critiquing existing efforts.

Introduction

Line 60-61: Considering this paper is regarding precision health mapping of diarrheal diseases, the paper “Variation in Childhood Diarrheal Morbidity and Mortality in Africa, 2000–2015” by Reiner et al (2015) should be considered.

Line 62: The authors should consider all mapping efforts from the Local Burden of Disease project which produce maps at the 5 x 5 km scale when discussing mapping efforts that are precise enough.

Line 75-82: While this may be true of many mapping efforts, this largely ignores many mapping papers that are fitted via GPS-tagged health data. The studies using such tagged data are not projecting national aggregates to lower spatial scales but rather fitting a Gaussian Process on point level data. Reiner et al (2015) and Mosser et al (2016) are two such examples.

The reviewer is correct in pointing this out, and we have better explained our reasoning in the current version. These studies do incorporate datasets with individuals or clusters who are GPS-tagged, similar to the cohort-level dataset in our own study. However, because they are designed to be representative at the national level, the spatial resolution of these datasets (the amount of data available per unit of space) is still much lower than the coverage of our dataset and may not be appropriate for predicting local

heterogeneities in the distribution of disease to inform decision making. We have added discussion of these studies to our manuscript [LINES 105 – 113]:

“For example, several recent studies analyzed the spatial patterns of disease and healthcare access across Africa, incorporating datasets that included GPS-tagged individuals and clusters (16–18) While these continent-level studies are comprehensive in their coverage across countries, the finest resolution dataset used for Madagascar (DHS 2008 surveys) had an average resolution of one cluster per approximately 1000 km². To improve local disease control, health managers in Madagascar make decisions at very small administrative levels (Fokontany, average size 34 km² in Madagascar) much finer than global or continent-wide studies are intended to address.”

Methods

Line 183: Why was the threshold of seven chosen?

This was originally chosen due to computational limitations. We have since gained access to a high performance computer and have removed this threshold. Because our final result is an averaged model that did include all variables (not just seven), this has not affected our results in a meaningful way.

Line 185: Why 2 delta AIC? What is its significance?

Typically, models that have a similar AIC regardless of the number of variables are considered to perform similarly because the AIC is a measure of model fit that already penalizes for model complexity. We chose models within 2 AIC so that the top model set consists of models whose performance in terms of model fit is similar to the top model, following established methods using information criteria (Burnham and Anderson 2002). We have added this reasoning to the manuscript [LINES 241-244].

Line 218: Was there any out-of-sample validation done on the model? One concern I have is that the model uses commune-level random effects for achieving an optimal fit over spatially structured effects. While this may fit the observed data well as you have a very flexible random effect to draw from for each location, this could perform poorly when you are trying to predict (or map) health outcomes in areas that have no data as your random effects are not following a structure. Considering health mapping is mainly used to predict health outcomes in unobserved locations this is a critical test.

In response to your concern, we have conducted out-of-sample tests on both of our analyses. The methods are described on LINES 237-240 and 286-290:

“To further assess the robustness of our findings, we also conducted out-of-sample testing. We included a randomly-sampled subset of 56 of the 80 clusters in our training dataset, and assessed the model’s performance predicting DD on the remaining 24 clusters.”

“We also assessed out-of-sample performance of this model by training the initial model on a random subset of 9 of 13 communes, stratified across the Pivot catchment area. Out-of-sample performance was then assessed on the remaining four communes, two of which were in the Pivot catchment area, accounting for fixed effects only. “

This has not influenced our results or conclusions significantly, as performance metrics remain very poor for the cohort analysis. However, by reducing our training data from 13 to 9 communes to train for the health system analysis, we have reduced our power and some covariates which were marginally significant are now marginally insignificant. This model also performs quite poorly predicting out of sample, further evidence that the model fails to adequately explain spatial variation in childhood diarrheal disease.

Results

Line 277-278: Has there been an assessment of collinearity or correlation between the covariates. If you fit a model with highly similar covariates the coefficient estimated for each of the highly correlated covariates can arbitrarily split the variance each of the covariate explains. Also, as noted above one should conduct some out of sample tests on both the cohort and monthly incidence model. This is simply telling you which covariates are the most important for this one set of observed data.

In response to comments from both reviewers, we chose to adjust how collinearity was addressed in our analysis. We had originally used the pre-scan method of selecting variables from a cluster of collinear variables based on their performance in univariate regressions of the response variable following Dormann et al. 2013. However, we do not want this to be interpreted as implying that the temperature and precipitation values at other lags are not predictive, as they were all highly correlated ($\rho > 0.9$) and so would have similar predictive capacities. Therefore, again following Dormann et al. 2013, we have chosen the most central variable of each group of climatic variables to represent the full range of temporal lags that were considered a priori. In the case of precipitation, this resulted in a lagged precipitation and cumulative precipitation value that were still correlated with $\rho > 0.7$. We therefore re-calculated the variable centrality measure across lagged and cumulative rainfall, resulting in one measure to represent precipitation patterns over the prior six months. This process was repeated separately for the model using cohort data and the model using health system data. The final variables chosen were temperature at a two-month lag and precipitation over the prior three months for the cohort data analysis and temperature at a one-month lag and cumulative rainfall over the prior three months for the health system analysis. This is described in the main text on LINES 195-205 and in the supplement.

In addition, we have included an out-of-sample test on both of our models (as described above).

Discussion

Overall: The discussion and finding regarding the importance of cultural factors such as the national holiday is very astute and an important finding for the field. Finding that cultural factors have a strong influence on disease state is an important point that should be focused on more.

Thank you. We have expanded this section to this point [LINES 556-560] and also mention it in the abstract.

Line 291-292: Is there a potential reason or mechanism that could explain why environmental variables are less influential than socio-demographic variables? Could it not be that the study scale is so small in scope geographically, that there is not enough spatial heterogeneity to allow for spatial effects or environmental effects to capture a significant amount of the variance?

Yes, we agree that this is a likely potential explanation for why a technique that works at national or multi-national scales is less effective at fine spatial scales. We have expanded on this point on LINES 511-532:

“Although the district spans an elevational gradient of 100 – 1400 m, its area is only approximately 45 x 125 km at its widest, which may not offer enough climate variation across space to observe a clear spatial trend or explain a significant amount of the variance in DD burdens. For precipitation especially, which is available at a 5 km resolution, there is little variation across clusters within a sampling period (Fig. 1). At this fine-scale, there may not be enough variation in the environment to explain differences in DD burden that would have been seen at national or multi-national scales, where environmental variation is greater. In contrast, socio-demographic variables are collected at the individual level and exhibit more spatial variation on a fine spatial scale.”

Line 297-299: I would be cautious to interpret regression coefficients as valid estimates of risk when the modeling framework has not been set up under a causal inference paradigm.

We have adjusted our language to be clearer that we do not intend to estimate risk in a causal way and refer to our findings associations, rather than risk, throughout the manuscript.

Line 314: This statement has many implications. An “improvement” in WASH is referring largely to sanitation facility type. Furthermore, it should be noted that the stated effect is only possible if there is an improvement to sanitation facilities without an accompanying improvement in hygiene behaviors.

We have changed this sentence to be more precise and dependent on this caveat [LINE 483]:

“When disease prevalence is high and WASH infrastructure coverage low, improvements to WASH infrastructure without accompanying improvements in behavior may counterintuitively increase DD exposure by concentrating exposure at sanitation points and increasing transmission due to human contact at these points, rather than water-borne transmission routes”

Appendix B

1785

The University of Georgia

®

January 14, 2021

Re: Manuscript No. RSPB-2020-2501

Dear Editor:

Please find enclosed with this letter our revised version of our manuscript “Socio-demographic, not environmental, risk factors explain fine-scale spatial patterns of diarrheal disease in Ifanadiana, rural Madagascar” which we resubmit for publication as a Research Article in *Proceedings of the Royal Society B*.

Your comments and those of the reviewers were insightful and informative and greatly enhanced the quality of our manuscript. In the following pages are our responses to each of the comments.

Reviewers’ comments are written in sans-serif font, with our replies directly following in *italics*. Revisions in the text are shown using Track Changes, and line numbers in the replies refer to this document. We hope that the revisions and our accompanying responses will be sufficient to make our manuscript suitable for publication in *Proceedings of the Royal Society B*.

I can be reached by email (mvevans@uga.edu), phone (703 725 9580), or post (Odum School of Ecology, University of Georgia, Athens GA 30602). Thank you for your consideration.

Sincerely,
Michelle Evans

We confirm that this manuscript is original and has not been published, is not in press, and is not under review elsewhere. All R code needed to reproduce the analysis will be deposited on Figshare upon acceptance and can currently be accessed via the following private link: <https://figshare.com/s/fc9cc786d9d6f9b7f84a>.

Please note that all line numbers correspond to the Track Changes document.

Comments to the Author(s).

The manuscript by Evans et al presents new insights about potential drivers of local-scale diarrhea incidence. While I recommend the manuscript to be published, I suggest few major issues that needs to be addressed by the authors.

Major issues

1. A first major issue with the manuscript is the interpretation and conclusion regarding holiday (national Independence Day) as a significant predictor. I agree that holidays may increase diarrhea cases, however, drawing a major conclusion based on a single holiday event maybe misleading. In addition, June 26 (or July used in the model) is within peak season of transmission. I suggest the authors to support this conclusion by looking at whether diarrhea cases also increase after other holidays.

We have chosen not to include other holidays in our analysis because national Independence Day was the only holiday specifically mentioned by local health care workers as a potential cause of increase in diarrheal disease incidence due to its unique characteristics, which we explain in further detail here. In rural Madagascar, where the vast majority of the population consists on poor subsistence farmers, national Independence Day is the major holiday happening after the main rice harvest period (March-May), where people have disposable cash to spend on celebrations. It is common practice for people in the community to pay contributions to a common fund in advance in order to buy livestock (mostly pigs) that are raised and killed specifically for national Independence Day so that meat is distributed among its members for the celebration. This, in addition to the numerous food stands set for this holiday results in an extraordinary consumption of meat products and fried food during and following National Independence Day that have no match the rest of the year. This sudden change in diet in combination with the lack of appropriate sanitary conditions to store these food products for days is locally hypothesized to drive the rise of diarrhea in the days and weeks that follow. While there are indeed other major holidays such as Christmas or Easter, these are mostly religious holidays where the main activity in rural areas consists on going to church, without other public celebrations or major changes in food consumption, as they happen before the harvest period.

We have clarified this in the methods section [lines 250-256]. Further, we ensure not to overstate our claim by implying that other holidays in Madagascar, besides the national holiday, are associated with increased DD incidence, although we do suggest that other studies should consider including culturally-relevant events when appropriate.

2. The authors failed to interpret the patterns of predicted disease risk for children across ages (Fig. 3C). A potential explanation might be due to rotavirus vaccination provided it has been introduced in the population. I suggest the authors examine these more closely and discuss the Fig. 3C.

In response to this, we have added an additional paragraph interpreting the patterns we found due to individual-level covariates, child sex and age [lines 374 – 385]:

In contrast, individual-level variables of child sex and age explained a large portion of the variation in DD prevalence. In particular, the non-linear trend in DD prevalence across age agrees with the results of continent-wide study of childhood DD in Africa (31). This

age-dependent pattern may be due to the implementation of the rotavirus immunization campaign in Madagascar since May 2014, which vaccinates children at two and four months of age (32). Indeed, there was a drop in cohort-wide prevalence from 16.18% to 8.41% from 2014 to 2018 following the start of this campaign; however, there was no evidence for a similar decrease in commune-level incidence rates across this time period (Fig. 4 A). As the immunization campaign continues, the planned collection of paired data on rotavirus vaccination status and DD in Ifanadiana will allow us to investigate this further.

Minor issues and typos

254 - sampling periods. There were between four and 30 children in each of the 80 clusters. I suggest authors to be consistent regarding four and 30.

We have chosen to follow APA guidelines for writing numbers (spelled out if less than 10, using Arabic numerals if more than 10), however we are happy to follow whatever style the journal prefers.

265 - then decreased with increasing age.. Delete one full stop.

Thank you for your sharp eye. We have corrected this typo.

Please also note that in response to page limitations imposed by the publisher, we have shortened the manuscript throughout and moved Table 1 to Table S1 in the supplement. However, this did not substantially change the overall narrative of the manuscript.